# Effect of Fabric Substrate and Introduction of Silk Fibroin on the Structural Color of Photonic Crystals

**DOI:** 10.3390/polym15173551

**Published:** 2023-08-26

**Authors:** Shu Yang, Hongming Xiang, Yingwen Wang, Kaikai Chen, Weihong Gao

**Affiliations:** School of Textiles and Fashion, Shanghai University of Engineering and Science, Shanghai 201620, China; 19851543881@163.com (H.X.); wyw1975193932@163.com (Y.W.); 39200002@sues.edu.cn (K.C.); gaoweihong@sues.edu.cn (W.G.)

**Keywords:** photonic crystals, structural color, fabric substrate, silk fibroin, color fastness

## Abstract

Monodispersed polystyrene (PS) particles were prepared and deposited onto various kinds of textile fabrics using a gravity sedimentation method. The monodispersed PS particles were self-assembled on fabrics to form a photonic crystal, which has an iridescent structural color. The structural color of fabrics was determined by the bandgaps of photonic crystals. Moreover, the effect of the fabric substrate, including the raw materials, base color, and fabric weave, etc., on the structural color of the photonic crystals was studied. Scanning electron microscopy and UV-vis spectrometry were adopted to characterize the structure and optical performance of photonic crystals. The results indicate that the silk fabric with a black base color and satin weave contribute to a bright and pure textile structural color. In order to solve the problem of low color fastness of the structural color on the fabric surface, silk fibroin (SF) was introduced to the PS microsphere solution. Results show that the addition of SF slightly affects the brightness of the structural color, while it has a certain reinforcing effect on the structural color fastness to rubbing and washing.

## 1. Introduction

Most colors of textiles are achieved by dyeing, which is called a chemical method. While many structural colors exist in nature, such as in butterfly wings, shells, and peacock feathers, structural color is generated by the interaction of light and the micro–nano-periodic structure [1], which is called the photonic crystal. It has the characteristics of high saturation, high brightness, a polarization effect, and unfading iridescence [2]. Therefore, the application of structural color to textiles has attracted scientists’ attention, hoping to either partially or completely replace the use of traditional dyes with the characteristics of high-level pollution and water consumption [3,4].

Photonic crystals are dielectric structural materials with photonic bandgaps formed by the periodic arrangement of materials with different dielectric constants in space. They were independently proposed by John [5] and Yablonovich [6] in 1987. Photonic crystals have many special physical properties and phenomena, such as a slow photon effect [7], a photonic bandgap [8], photon localization [9], a super prism effect [10], a negative refraction effect [11], etc. According to the periodic spatial distribution of refractive index changes, they can be divided into three categories [12,13]: one-dimensional, two-dimensional, and three-dimensional ones. One-dimensional photonic crystals often produce structural coloration through film interference and grating diffraction. A two-dimensional photonic crystal is a material in which the dielectric constant of a medium is periodically arranged in two directions in the spatial plane and uniformly distributed in the vertical direction of the plane. A two-dimensional photonic crystal has a very small geometric size and photonic bandgap, and it has the characteristics of reduced transmission loss, a self-imaging effect, and a slow light effect. A three-dimensional photonic crystal is a material in which the dielectric constant of a medium is periodically arranged in all three directions in space and can produce structural coloration when it interacts with visible light. Based on the characteristics of photonic crystals, when the bandgap of photonic crystals is within the visible light range, structural color can be achieved [14]. The photonic bandgap of colloidal crystals is greatly influenced by the size and arrangement of colloidal particles [15].

Many researchers have conducted in-depth research on application of the structural color to fabric surfaces. Zhang [16] focused on the preparation of photonic crystal structural coloration fibers, summarizing the research and application progress of multi-functional structural coloration fibers. Shao conducted research on SiO_2_ photonic crystals on cotton fabrics [17], polyester fabrics [18], and yarns [19]. Shi [20] coated polydopamine (PDA) on cotton fabrics, and then self-assembled photonic crystals, which greatly improved the color fastness and color saturation of structured dyed fabrics. Liu [21] prepared bright-iridescence structural colors using vertical deposition on a cotton fabric. Yuan [22] used a white polyester fabric to prepare structural coloration fabrics and an Ag/TiO_2_ composite film via magnetron sputtering.

In recent years, research on self-assembly technology of photonic crystals has also received great attention [23,24,25,26,27,28]. The fabrication methods of photonic crystals include vertical deposition, gravity deposition, electrophoretic deposition, ink-jet printing, etc. 

PS (polystyrene) microspheres are a type of colloidal particles commonly used in the fabrication of photonic crystals due to their perfectly spherical shape, smooth surface, and uniform particle size [29,30,31,32,33]. In general, the artificial photonic crystals are fabricated from a binary system, which is made from PS microspheres and air. However, photonic crystals made from this binary system usually have weak bonding with fabrics. Hence, we propose the use of a soft material such as silk fibroin (SF) (acquired from silkworm silk) to replace the air and form a complex with the PS microspheres. Compared with air, SF has better elasticity and moisture, so it can enhance the bonding between photonic crystals and fabrics. A silk fibroin solution can be obtained by removing the sericin protein and subsequently dissolving fibroin in protein denaturants (e.g., lithium bromide solution) [34].

This paper aims to determine the optimal fabric substrate to obtain the desired structural color of textile fabrics. The influence of fabric raw materials, base colors, and fabric weaves on the structural color are examined, respectively. Meanwhile, in order to solve the low bonding strength between photonic crystals and fabrics, silk fibroin is added to the colloidal solution, and then deposited on fabrics. The color fastness to rubbing and washing is characterized.

## 2. Materials

The fabrics made of different raw materials (cotton, silk, and wool), different colors (white, red, and black), and different fabric weaves (plain, twill, and satin) were purchased from the Shanghai Fabric Market (Shanghai, China). Bombys mori silkworm cocoons were provided by Guangxi Sericulture Technology Co., Ltd. (Nanning, China). The monodispersed polystyrene (PS) microspheres were made in our laboratory. Styrene (St, analytically pure), potassium persulfate (KPS, analytically pure), sodium bicarbonate, and lithium bromide were purchased from Shanghai Aladdin Technology Co., Ltd.(Shanghai, China). High-purity deionized water (18.2 MΩ), produced by a Millipore Milli-Q system (Burlington, MA, USA) (0.22 um), was used for preparing buffers, which were used as solvents for mixed solutions. 

## 3. Experiments

### 3.1. Synthesis of PS Microspheres

The soap-free lotion polymerization of the PS microspheres was carried out in a four-necked, jacketed glass reactor equipped with a nitrogen bubbler, a top D-shaped mechanical agitator, and a condenser. Firstly, deionized water (100 mL) was introduced into a glass reactor that was purged with nitrogen gas for 10 min. Then, styrene was added to the reactor, and the mixture was vigorously stirred in a nitrogen atmosphere at 330 rpm for 10 min, while the temperature was raised to 70 °C. After that, potassium persulfate dissolved in a small amount of water was added, and the polymerization reaction was carried out at 70 °C under nitrogen protection for 20 h. After cooling it to room temperature, the filtrate was saved for the following experiments. PS microspheres with particle sizes of 260 nm ± 5 nm were synthesized. 

### 3.2. Self-Assembly of Photonic Crystals

The fabrics were treated with ultrasonication in deionized water and dried before use to ensure a clean surface. The gravity deposition method was used to form photonic crystals on fabrics (Figure 1). PS colloidal suspensions were dispersed in ultrasound for 50 min, and then deposited dropwise onto the fabric surface. The photonic crystals were self-assembled in a vacuum oven at 60 °C with a relative humidity of 50% for 5 h.

### 3.3. Preparation of Silk Fibroin Solutions

Silk fibers were degummed in a boiling aqueous solution of 0.5% (*w*/*w*) NaHCO_3_ for 30 min, twice, with frequent stirring. After that, the degummed silks were washed with deionized water 5 times. The regenerated silk fibroin (SF) was acquired by dissolving the degummed silks into a 9.3 M LiBr solution for 4 h at 60 °C, and then extracting LiBr from the SF solution via a dialysis cassette (Solarbio, molecular weight cut-off: 3500) for 2 days. 

Fluorescein isothiocyanate (FITC)-labeled SF was synthesized based on the covalent conjugation of isothiocyanate group pf FITC and the amino group of SF. Briefly, a 1 mL Na_2_CO_3_ (0.5 mM) solution was added to 10 mL of a 70 mg/mL SF solution, and then 7 mg of FITC in 1.4 mL of DMSO was added to the as-prepared SF solution. Then, the mixed solution was slowly stirred for 2 h in a dark room at 25 °C. Finally, the FITC-labeled SF solution was obtained by dialysis in the dark for 6 h to remove unconjugated FITC. Afterwards, 200 μL of 1% (*w*/*v*) PS particles suspension was centrifuged at 2500 rpm, and the PS particles were then re-dispersed in 1 mL of a 5 mg/mL FITC-labeled SF solution in the dark [35].

### 3.4. Color Fastness Measurements

The color fastness to rubbing was measured according to GB/T 3920—2008 [36]. The samples were cut into a 50 mm × 140 mm size and fixed at the lower testing position along the direction of the friction head’s round-trip path. At the same time, about (9 ± 0.2) N of pressure was applied to the friction head, and the friction head movement was adjusted to (104 ± 3) mm, 1 cycle per second, for a total of 100 cycles. After rubbing, the reflective spectra of samples were measured.

The color fastness to washing was measured according to GB/T 3921—2008 [37]. The samples were cut into a 100 mm × 40 mm size, then washed in a 1 g/L soap solution in a 40 °C water bath, taken out, and dried in an oven. The reflective spectra of samples were measured after soaping for 30 min, 60 min, 90 min, and 120 min, respectively.

### 3.5. Characterization

The microstructures of photonic crystals were characterized by scanning electron microscopy (Hitachi SU70, Tokyo, Japan) at 1.0 kV, and the samples were not coated with gold. The size distribution and mono-dispersion of PS microspheres were characterized by DLS (NanoBrook Omni Brookhaven, Schenectady, NY, USA). The reflective spectrum of photonic crystals was measured with a UV-vis spectrometer (USB-2000, Ocean Optics, Dunedin, FL, USA). The fluorescence images of FITC-labeled, SF-incubated PS were obtained with a Leica TCS SP8 confocal laser scanning microscope (Wetzlar, Germany).

## 4. Results and Discussions

### 4.1. Dispersion of PS Microspheres

Figure 2 shows the PDI curve of the PS microspheres. It can be seen from the figure that the PDI of the synthesized PS microspheres was less than 0.1, which proves that the particle size of PS microspheres was uniform.

### 4.2. Effect of Raw Materials

The self-assembled PS microspheres with 260 nm and 300% owf were deposited on plain cotton, silk, and wool fabrics via the gravity sedimentation method. Figure 3a–c show the fabrics made from different raw materials, but with the same plain weave and black color. The fineness of warp and weft filament was 40 s × 40 s, and the fabric density was 128/10 cm × 68/10 cm.

The structural color on the surface of the silk fabric (Figure 3e) exhibited a bright and uniform color, and obvious iridescence, which can be proven by the reflection spectrum curves (Figure 3h).

In the spectrum curve, the peak position corresponding to the wavelength (nm) represents the coloring phase. While in photonic crystals, the coloring phase is determined by the position of the bandgap, which can be adjusted by the size of the colloidal microspheres. The position of the reflection peak is considered the wavelength of the photonic bandgap, where a certain wavelength range of electromagnetic incident light is forbidden to propagate. The peak positions in Figure 3g–i were nearly the same (576 nm ± 5 nm, 578 nm ± 5 nm, 574 nm ± 5 nm) as the microspheres deposited with the same size (260 nm ± 5 nm). This means that the bandgap of photonic crystal falls into 576 nm, and the incident light of this specific wavelength cannot be allowed to propagate but can be reflected, which is why the fabrics seem to be green to blue, which is the color of visible light near 576 nm.

The peak height in the spectrum curve represents the brightness of the color. The higher the peak, the brighter the color appears. Photonic crystal on silk fabric had the brightest structural color, whose peak height was 29.312 ± 0.003 au, compared to that on cotton (23.556 ± 0.002) and wool (27.468 ± 0.004).

The peak width (nm) represents the purity of the color: the smaller the peak width, the better the purity of the color. The photonic crystal on silk had the narrowest peak (125.452 ± 0.02), compared to that on cotton (189.364 ± 0.01) and wool (132.246 ± 0.02).

Figure 4 shows SEM images of the photonic crystals self-assembled on fabrics with different raw materials. Figure 4a shows the macrostructure of silk fabric after the deposition of PS particles. It can be clearly seen that the surface of the fabric is very rough, which is due to the deposition of PS microspheres. The arrangement of microspheres on the silk fabric (in Figure 4b) is more regular than those on the cotton and wool fabrics (Figure 4c,d).

Despite having the same plain weave, due to the natural twisting structure of cotton fibers, and the scale structure on the surface of wool fibers, the surface of cotton and wool fabrics was not as flat as that of silk fabrics, resulting in many more defects in the photonic crystals fabricated on cotton and wool fabrics. Such obvious cracks in the photonic crystal can not only decrease the reflectivity of the light in the bandgap but can also scatter the nearby lights outside the bandgap wavelength.

The phenomenon of photonic crystals with an ordered structure can be explained by Bragg’s law. When the bandgap of photonic crystal falls into the visible light region, the photonic crystals exhibit structural colors. The different effects of structural colors were produced by different self-assembled photonic crystals. It is well known that for a perfect photonic crystal, the incident light could be strongly reflected at the photonic bandgap, while in the other wavelengths the lights will have very high transmittance. The regular arrangement of microspheres on silk fabric (Figure 4b) will lead to a more obvious effect of the structural color, which means that the color may be brighter and much more angle-dependent.

### 4.3. Effect of Fabric Base Colors

The colloidal PS microsphere solution (260 nm) was deposited on fabrics with owf 300% using the gravity sedimentation method, mainly considering black, red, and white fabric base colors. Figure 5a–c show fabrics with different base colors. All the fabrics were made of silk with a satin weave, and the fineness of warp and weft filament was 40 s × 40 s, while the fabric density was 128/10 cm × 68/10 cm.

Figure 5d–f show the structural color on silk fabrics with different base colors. The structural color effect on the black surface appears the best, followed by red silk, and then white silk. To our naked eyes, the photonic crystal deposited on the white silk fabric seems almost white, with no special structural color, but the color is very bright, as shown in Figure 5d. The photonic crystal deposited on the red silk fabric appears orange, as shown in Figure 5e, and the photonic crystal deposited on the black silk fabric appears green, as shown in Figure 5f—this is the closest to the structural color of the photonic crystal bandgap.

The results of spectrum curves in Figure 5g–i are consistent with the apparent structural color effect.

In Figure 5g, the white silk fabric has a high reflectance value of near 90% in the whole visible wavelengths, which indicates that most of the incident light is reflected and less is absorbed. However, there is still a small reflection peak near wavelength 578 nm, which is the bandgap of the photonic crystal fabricated by PS colloidal microspheres on the white silk fabrics.

For the photonic crystal on the red silk fabric (Figure 5h), the reflective curve shows two peaks at 576 nm and 691 nm. The peak at 576 nm is produced by the bandgap of the photonic crystal, and the peak at 691 nm comes from the original base color of the silk fabric. When the red fabric was deposited by the photonic crystal whose bandgap is near the green light wavelength, the fabric seemed to be orange, which is mixed by red and green light.

The peak height of the photonic crystal on the black surface (Figure 5i) is the highest (31.554 ± 0.001), which means that the structural color is the brightest and the peak width is the narrowest (127.119 ± 0.03), which indicates that the purity of structural coloration on the black surface of the silk fabric is the best.

The same PS colloidal microspheres deposited on silk fabrics with different base colors show different structural colors. This is because the observed light not only contains the reflective wavelength caused by the photonic bandgap, but also the wavelengths beyond the photonic bandgap, which decrease the purity of the structural color [2]. When the base color of fabric is black, the black color will absorb the redundant transmitted and scattered light beyond the photonic bandgap, and hence improve the effect of the structural color corresponding to photonic crystals. Conversely, when the white silk fabric is used as a substrate material, it can not only reflect the certain wavelength of visible light that is forbidden to propagate by the photonic crystals but can also reflect the transmitted and scattered light beyond the photonic bandgap wavelength. This will dilute the structural color from the selective wavelength of the photonic crystal bandgap. That is why the same colloidal microspheres on different fabrics show different structural color effects and reflectance spectra.

### 4.4. Effect of Fabric Weaves

The colloidal microsphere solution (260 nm, 300% owf) was deposited on silk fabrics with different weaves, including plain weave, twill weave, and satin weave, using the gravity sedimentation method. Figure 6a–c show the black silk fabrics with different weaves of plain, twill, and satin, respectively. The fineness of warp and weft filament was 40 s × 40 s, and the fabric density was 128/10 cm × 68/10 cm.

The hues of structural colors on fabrics with different weaves appeared green, but the brightness of them was different, which can be seen from Figure 6d–f. To naked eyes, the effect of the structural color on the silk fabric with the satin weave appeared the best, being the brightest and purest.

From Figure 6g–i, it can be seen that the peak width of the satin fabric (127.119 ± 0.03) is smaller than that of the plain weave and twill silk fabrics, which means the purity of structural coloration on the surface of the satin silk fabric is the best. The peak height of satin fabric (31.554 ± 0.001) is higher than those of the other ones, which means the satin fabric has the optimal brightness.

The texture points of the plain and twill fabrics are densely distributed, and the fabric is relatively stiff, while the satin fabric has fewer texture points, longer floating lines, and a smooth surface, making it more suitable for the optimum structural color effect. From the SEM images of photonic crystals on different weaves, it can be seen that the photonic crystal self-assembled on the satin weave in Figure 6l has the most regular structure and the least defects. Relatively, the structures of the photonic crystals on the plain weave in Figure 6j and the twill weave in Figure 6k show more cracks and defects, and this is due to the fabrics with plain and twill weaves having more interweaving points and undulations on the surface, which brings the self-assembly of photonic crystals more difficulties.

### 4.5. Effect of Silk Fibroin on Color Fastness

In this study, solutions of PS microspheres and silk fibroin were mixed to form a complex solution. During the synthetic process, the PS microspheres were grafted using the carboxyl group from acrylic acid, while silk fibroin is a protein which contains the amino and carboxyl groups. When these two solutions were mixed, strong hydrogen bonds formed between the carboxyl group on the PS microspheres and the amino groups on the SF. This supplied high strength for combining these two molecules. Due to the carboxyl group on the surface, PS microspheres are hydrophilic. However, SF possesses both hydrophilic and hydrophobic groups and is amphiphilic. Therefore, the hydrophilic groups of SF connect with PS, and the hydrophobic ends coil inwards during the first stage. Later, the hydrophobic ends try to escape from the water and stretch to the solution–air interface to make the system more stable by decreasing the free energy. Simultaneously, the whole complex gets pulled together at the interface. Finally, the hydrophobic parts of SF spread out on the surface, and the hydrophilic parts accompanying the PS spheres assemble at the interface. Thus, the hydrophilic parts of the SF molecule form a glue-like material among the PS spheres [35].

For better validation of the combination of silk fibroin and PS spheres, confocal laser scanning microscopy (Figure 7a) and SEM (Figure 7b) were adopted, and the grafting of SF was traced on the PS surfaces. Fluorescein isothiocyanate (FITC) showed covalent bonding with the SF and labeled specific molecules. After incubating the PS particles within the FITC-labeled SF solution for 30 min, they were harvested through centrifugation. The detectable SF molecules labeled by fluorescence were found to aggregate, surrounding the microspheres after incubation. Besides, according to the SEM images regarding PS subjected to incubation within SF solutions, SF molecules accumulated around the PS particles, and connected them, acting as glue.

The addition of SF to the PS solution may affect the structural color caused by photonic crystals. Figure 8 shows spectrum curves of photonic crystals on silk satin fabrics. Compared with that made from the pure PS solution, the peak height of that from the PS-SF solution dropped by 13.8%. These results prove that silk fibroin would slightly affect the structural color of photonic crystals. The addition of SF changed the shape and size of the microspheres, and then the regularity of the photonic crystal decreased. At the same time, fibroin protein covered the PS particles, affecting light scattering and the brightness of the structural color.

The spectrum curves of photonic crystals before and after rubbing 100 times, fabricated from the pure PS and PS-SF solutions, were measured and shown in Figure 9. For the photonic crystals made from pure PS spheres, the peak height of the spectrum curve dropped by 72.7% after rubbing, as shown in Figure 9a, while that of the photonic crystals made from the PS-SF mixed solution only dropped by 57.2%, as shown in Figure 9b. This indicates that the addition of SF has a certain reinforcing effect on the structural color fastness to rubbing on the fabric surface. This is due to the glue-like SF molecules between the fabric and the photonic crystals having a certain degree of viscosity, which makes the bonding of PS microspheres to the fabric surface much more firm.

The spectrum curves of photonic crystals before and after washing for 30 min, 60 min, 90 min, and 120 min were measured and shown in Figure 10. Figure 10a shows the spectrum curve of the photonic crystals constructed from the pure PS microsphere solution. The peak height decreased by about 50% after 30 min of washing, and then continuously decreased with the increasing washing time. The biggest decrease occurred in the first 30 min. Figure 10b shows that for the photonic crystals constructed from the PS-SF solution, the peak height only decreased by 15% after washing for 30 min, and dropped by 50% after 120 min of washing. This indicates that the addition of SF significantly improves the structural color fastness to washing. The reason for this is that the swelling and adhesion effect of SF in water makes the bonding between photonic crystals and the fabrics more tight.

## 5. Conclusions

Monodispersed polystyrene particles were prepared using soap-free emulsion polymerization and deposited onto various kinds of textile fabrics using the gravity sedimentation method. The monodispersed PS particles were self-assembled on the fabrics to form photonic crystals, which have an iridescent structural color. The structural color of the photonic crystals on fabrics is determined based on the bandgaps and can be affected by the fabric surface. Scanning electron microscopy (SEM) observation and UV-vis spectrometry results indicated that fabrics made from silk, with a black base color and a satin weave, contribute to a bright and colorful structural color.

Silk fibroin was introduced to the PS microsphere solution to solve the problem of low color fastness of the structural color to the fabric surface. The addition of SF slightly affected the structural color of photonic crystals, while it had a certain reinforcing effect on the structural color fastness to rubbing, and significantly improved the structural color fastness to washing.

The application of the structural color to textiles would take the place of chemical dyes. Future research should be conducted on the large-scale industrialization of the self-assembly of photonic crystals and further enhancement of binding fastness to textiles.

## Figures and Tables

**Figure 1 polymers-15-03551-f001:**
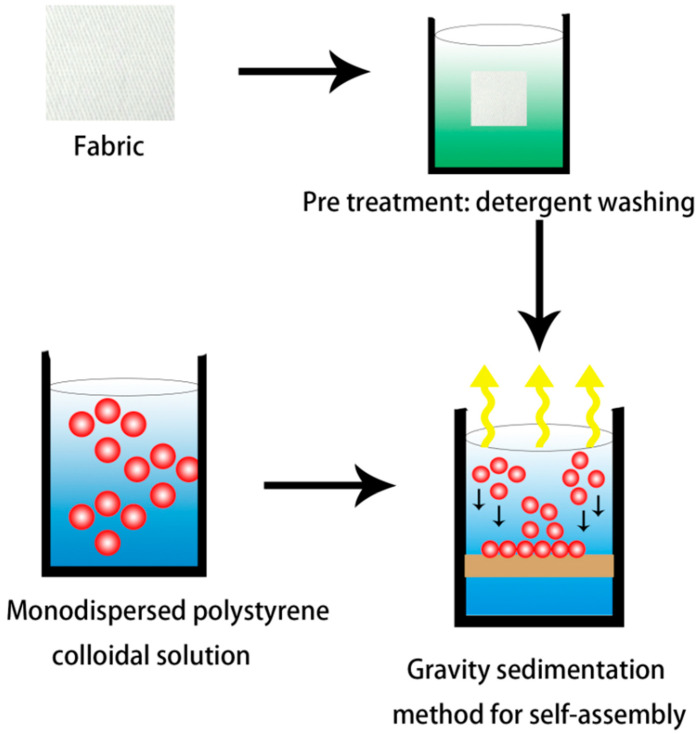
Fabrication of PS photonic crystals on fabrics.

**Figure 2 polymers-15-03551-f002:**
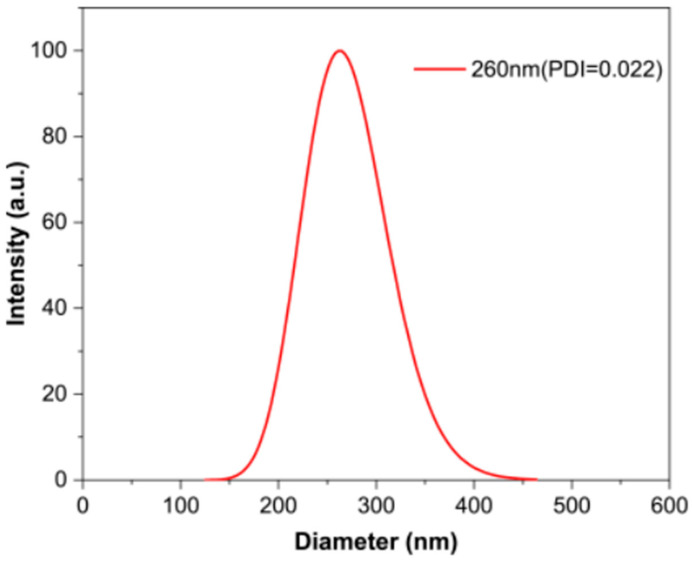
Size distribution of PS microspheres.

**Figure 3 polymers-15-03551-f003:**
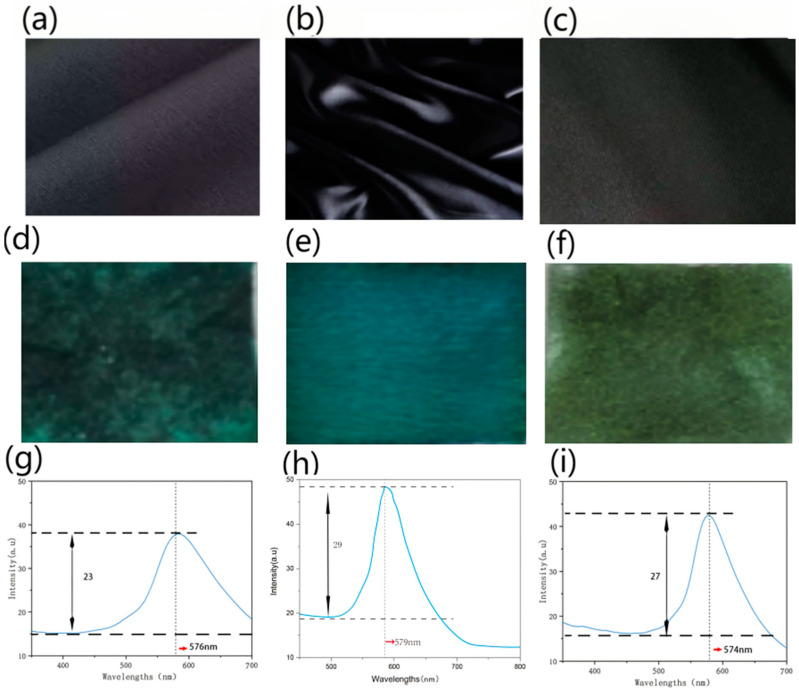
The structural color of fabrics with different raw materials of (**a**) cotton, (**b**) silk, and (**c**) wool. (**d**–**f**) Images of fabrics after deposition, and (**g**–**i**) reflection spectrum curves of the photonic crystals on different raw fabric materials.

**Figure 4 polymers-15-03551-f004:**
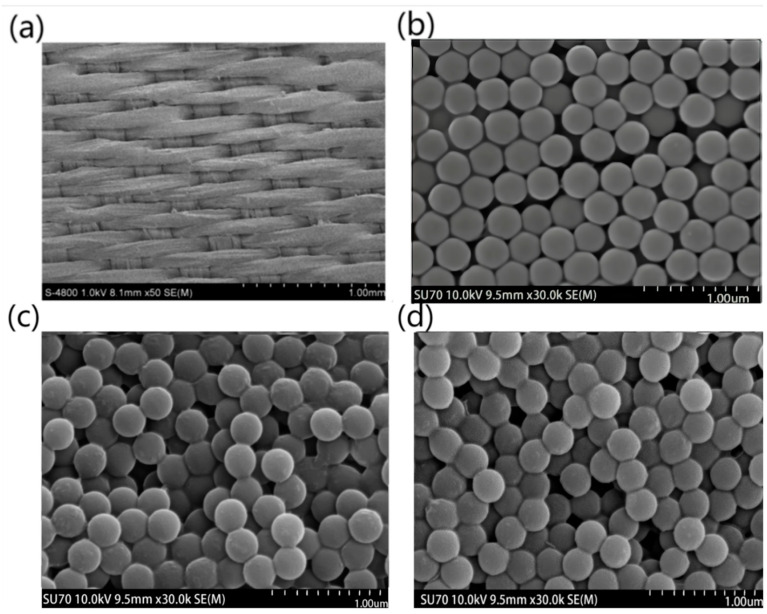
SEM images of photonic crystals self-assembled on different fabrics of (**a**, **b**) silk, (**c**) cotton, and (**d**) wool.

**Figure 5 polymers-15-03551-f005:**
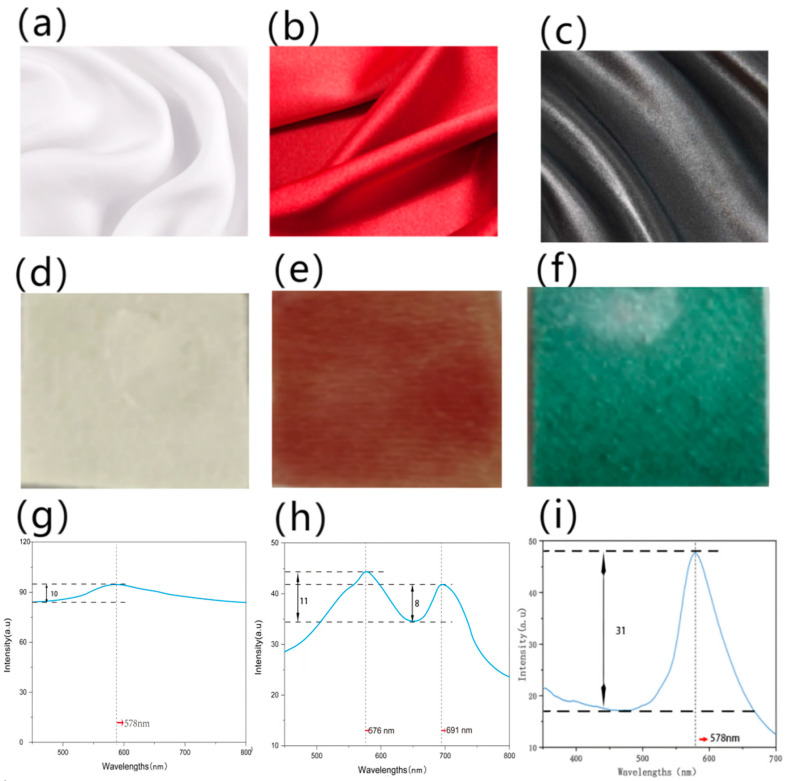
The structural color of silk fabrics with different base colors: (**a**) white, (**b**) red, and (**c**) black. (**d**–**f**) Images of fabrics after deposition and (**g**–**i**) reflectance spectrum of photonic crystals on fabric surfaces in different base colors.

**Figure 6 polymers-15-03551-f006:**
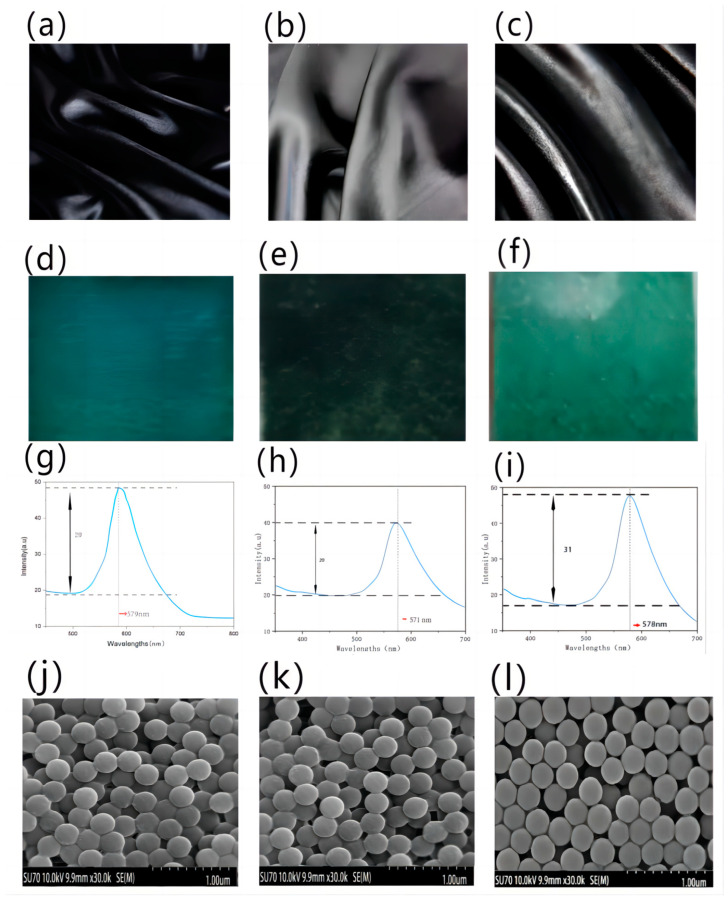
The structural color of silk fabrics with different weaves: (**a**) plain, (**b**) twill, and (**c**) satin. (**d**,**e**,**f**) Images of fabrics after deposition, (**g**,**h**,**i**) reflectance spectrum curves, and (**j**,**k**,**l**) SEM images of photonic crystals on silk fabrics with different weaves.

**Figure 7 polymers-15-03551-f007:**
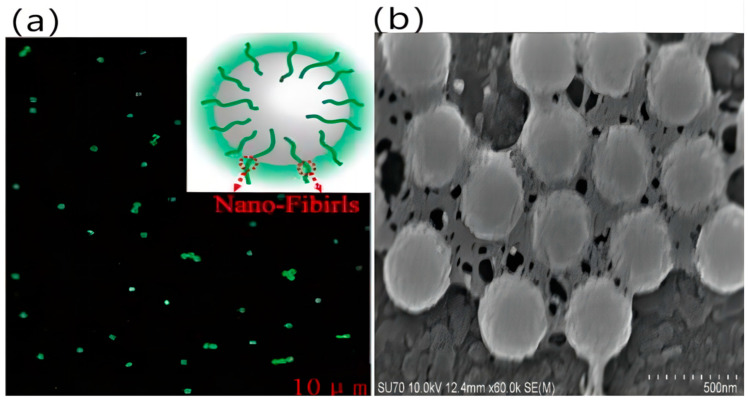
(**a**) Confocal laser scanning microscopy image of PS particles, which were incubated in FTIC-labeled SF solution for 30 min. (**b**) SEM images of PS particles incubated in SF solutions for 30 min.

**Figure 8 polymers-15-03551-f008:**
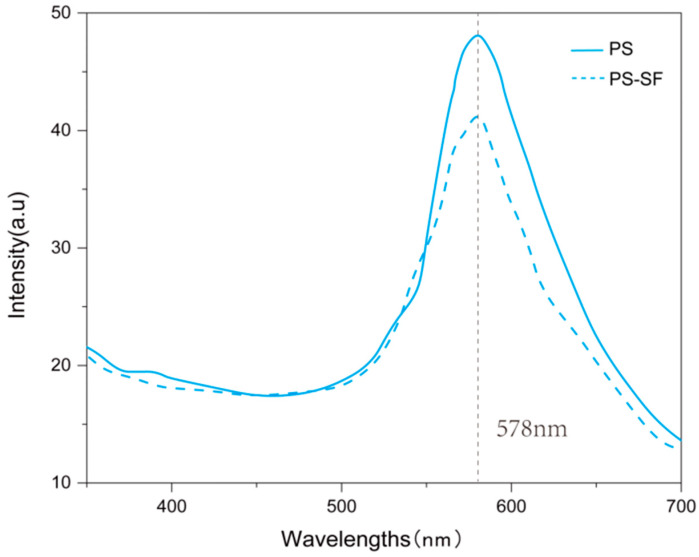
Spectrum curves of photonic crystals fabricated from PS and PS-SF solutions.

**Figure 9 polymers-15-03551-f009:**
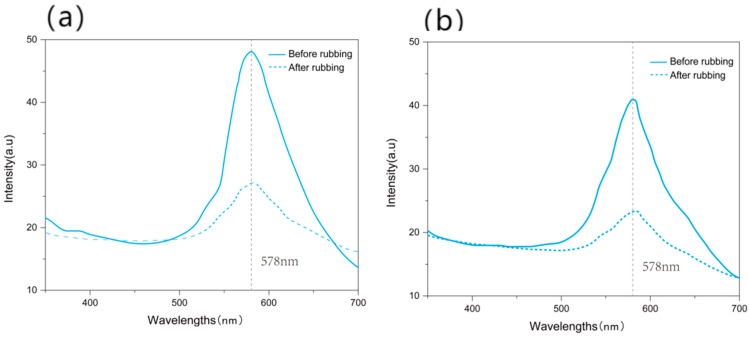
Spectrum curves of photonic crystals fabricated from (**a**) pure PS and (**b**) PS-SF solutions before and after rubbing 100 times.

**Figure 10 polymers-15-03551-f010:**
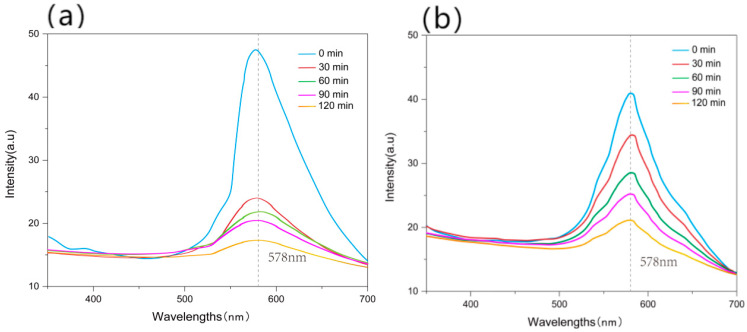
Spectrum curves of photonic crystals fabricated from (**a**) pure PS and (**b**) PS-SF solutions before and after washing.

## Data Availability

Not applicable.

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
