# Peer review of "Effect of Fabric Substrate and Introduction of Silk Fibroin on the Structural Color of Photonic Crystals"

_polymers, 2023, doi:10.3390/polym15173551_

Round 1
Reviewer 1 Report (Previous Reviewer 4)
It is revised as per my suggestions.
I therefore recommend accepting for publication.
Minor editing needed.
Reviewer 2 Report (Previous Reviewer 3)
The revised manuscript was significantly improved.
Reviewer 3 Report (Previous Reviewer 2)
Dear Authors,
You have tkaen into account the different comments made previously. So, the modifications and the addings made improved deeply the manuscript.
Reviewer 4 Report (Previous Reviewer 1)
Dear Author
Thank you answering the comments and questions
your article can be accepted for publication
This manuscript is a resubmission of an earlier submission. The following is a list of the peer review reports and author responses from that submission.
Round 1
Reviewer 1 Report
Dear Author
Thank you for this idea, but I need to know the field you target by this work, is this product present in the market? We need comparing with keeping in mind the aim of its uses.
why you apply on the three different type of fabric silk, cotton and polyester? why you investigate the weaving structure only on silk? if you suggest it will affect the property, you should investigate also the effect on the other structure fabric type
Reviewer 2 Report
Dear Authors,
The main idea to develop alternative coloration mode for textiles is very interesting to overcome the ecological problems of conventional deying. This can lead to develop more sustainable textile fabrics.
As mentioned in the inroduction structural coloration is formed by the interaction of light and micro-nano structure. However the structure of textiles fabrics are not sufficiently studied in this paper. For example in page 3 you mention that the substrate is flat. I recommend to add topography or rugosity measurements to support your hypotheses. You should also add more details about the yarns used for the fabrics, i.e: staple yarns or filament yarn, linear density (Mn or dTex), also more details about the fabrics (not only the structure : plain, satin, twill), i.e: number of weft yarn per inch, or warp yarn per inch.
The arrangement of photonic crystal is of a prime importance to expalin the coloration. Unfortunately there is only one SEM image for one sample made on silk fabric. The other images should be added and deeply analysed to explain the differences observed in term of coloration.
The comments like ("longer floating lines, ...surface" (page 11, line 25) should be supported by measurements.
Reviewer 3 Report
The authors explored the effects of particle size, fabric structure, base color, and material types on the structural color generation. The paper needs to be improved in the following aspects:
1. In line 103 and 153, the authors mentioned using P (St-MMA-AA). However, only synthesis of PS particle was described. This makes the reader wonder what is the exact material, and what is the formulation (concentration of the monomer, initiator etc.)
2. The synthesis procedure described in section 3.1 is insufficient, methods used for controlling the size of the particle was unclear. The methods used for particle size measurement was not stated. Size distribution needs to be given in order to claim monodisperse.
3. Table 2 and Figure 3 should be numbered with easy to follow order so that it will be easy to read the pictures. Line 195-201 were basically all spent describing samples. This can be saved if you put better number or legend in the Figure.
4. Figure 2 (b). The magnification was clearly indicated on the SEM micrograph as 30k, much larger than the 5000x the author wrote.
5. Line 229-231, the statement does not match the table result.
6. Was experiment result present in Figure 3 done on all white fabrics? If so, why Figure 5 a-c does not show good color while Figure 3 b e h which are also silk showed blue and green color?
Reviewer 4 Report
The paper is of significance and of interest. However, it needs significant improvements before publication.
1. The figures 1, 4, 6 and 8 are of poor resolution and clarity. They must be improved.
2. The images of blank fabrics should be given so as to evaluate their color and brightness. The effect of weave on the control sample should be determined.
3. What about color measurement and K/S values? why were they not presented?
4. How can the color differences be significant? Is there any quantitative method to evaluate them?
5. After quantification of measurement data, they should be trated statistically to show the significance of the values (standard deviation, error %, coefficient of variation etc.
6. The conclusion part is too brief and generic. It should include the quantitative findings of the research.
7. The paper needs a rewriting to enhance its scientific value.
8. In abstract it starts as PS particle....what is PS particle? All abbreviations should be expanded.
I suggest a major revision.
Yes language editing needed.
Round 2
Reviewer 2 Report
Dear Authors,
as mentioned previously the idea to color textile without using deying treatment is very interesting. However you should more explain how the different organization of PS particles on different substrates can explain the results.
Author Response
Thank you for your kind advice.
We realize that showing organization of PS particles on different substrates is very important to explain the different effect of structural color.
We have added the SEM images of photonic crystals on different fabrics in Fig.4, and also added relevant explanations in manuscript which is highlighted.

Reviewer 3 Report
The authors have improved the manuscript.
Author Response
Thank you for taking the time to review our manuscript.
Reviewer 4 Report
The paper is revised.
However, still the first mention of PS in the abstract should be expanded as Polystyrene.
The pictures of fabrics shown in Fig.s 2, 7 and 10 should be taken with a much higher resolution camera. At present they are of very poor quality.
Still the abstract and conclusion need to be expanded.
English editing neeeded.
Author Response
Thank you for your suggestion.
We have rewritten the abstract and conclusion, and used English editing service to improve the language.
And also we have updated clearer pictures in Fig.s 2, 7 and 10.